# Ameliorative Effect of *Sipunculus nudus* Hydrolysate on Cisplatin-Induced Nephrotoxicity by Mitigating Oxidative Stress, Inflammation and Apoptosis

**DOI:** 10.3390/md23030100

**Published:** 2025-02-24

**Authors:** Susu Tao, Yi Qi, Jialong Gao, Huafang Yuan, Ruimin Wang, Xiaoqin Shen, Gang Wei, Zhilan Peng

**Affiliations:** 1School of Ocean and Tropical Medicine, Guangdong Medical University, Zhanjiang 524023, China; t463262157@163.com (S.T.); qiyi7272@gdmu.edu.cn (Y.Q.); wgang@gdmu.edu.cn (G.W.); 2College of Food Science & Technology, Guangdong Provincial Key Laboratory of Aquatic Products Processing and Safety, Zhanjiang 524088, China; gaojl@gdou.edu.cn; 3The Marine Biomedical Research Institute of Guangdong Zhanjiang, Zhanjiang 524023, China; 18085434415@163.com (H.Y.); ruimin@gdmu.edu.cn (R.W.); sxq11192021@163.com (X.S.)

**Keywords:** *Sipunculus nudus*, cisplatin, nephrotoxicity, hydrolysate, oxidative stress, inflammation, apoptosis

## Abstract

The present study investigated the protective effects and possible mechanisms of an ultrafiltration fraction of *Sipunculus nudus* hydrolysate (UFSH) on cisplatin-induced kidney damage in a mouse model. The results showed that UFSH significantly attenuated cisplatin-induced nephrotoxicity by inhibiting increases in blood urea nitrogen (BUN) and serum creatinine (SCr). Additionally, UFSH treatment significantly alleviated cisplatin-induced renal histopathological changes, such as significant dilation of renal tubules, cast formation, and tubular cell necrosis, as well as tubulointerstitial fibrosis. Moreover, UFSH decreased cisplatin-induced oxidative stress by increasing the activities of antioxidant enzymes SOD and GSH-Px, while reducing the malondialdehyde (MDA) level in the kidney. Furthermore, UFSH significantly inhibited cisplatin-induced increases in inflammatory cytokines, including Interleukin 1-beta (IL-1β), Interleukin-6 (IL-6), and Tumor necrosis factor-alpha (TNF-α). Western blotting revealed that UFSH inhibited the phosphorylation of the inflammation-associated MAPK/NF-κB signaling pathway, lowered the expression of the apoptosis-related protein Bax, and reversed the reduction in the anti-apoptotic Bcl-2 protein. This investigation demonstrated that UFSH can ameliorate cisplatin-induced nephrotoxicity by mitigating oxidative stress, inflammation, and apoptosis.

## 1. Introduction

Cisplatin is a kind of broad-spectrum anticancer drug that is widely used in the treatment of many solid tumors. However, the adverse reactions of cisplatin limit its clinical application, with nephrotoxicity being a common and serious adverse reaction [1,2]. Although cisplatin treatment has considerable effect for cancer therapy, 20–30% of patients develop acute kidney injury (AKI) after cisplatin application [3,4]. Patients with AKI are more likely to progress to chronic kidney disease (CKD), which in turn increases the risk of cancer-related mortality [5,6]. The nephrotoxicity of cisplatin is cumulative and dose-dependent, necessitating dose reduction or discontinuation [7]. Despite huge efforts that have been made to find less toxic but equally effective alternatives over the decades, cisplatin remains a standard part of cancer treatment regiments [8]. Therefore, the prevention and treatment of cisplatin-induced nephrotoxicity is still of great importance.

Currently, various natural compounds [7], plant extracts [9,10], and protein hydrolysates [11,12] have been reported to exert protective effects on cisplatin-induced nephrotoxicity. Particularly, bioactive peptides from marine organs have gained increasing attention from researchers in recent years attention by researchers in recent years. *Sipunculus nudus* is an important economic invertebrate widely found in the sand and mud bottom of coastal areas. *Sipunculus nudus* possesses properties of delicious taste, rich nutrition, a high content of essential amino acids, and a relatively complete range of species. Traditionally, *Sipunculus nudus* has been used by citizens around the coast of China to treat chronic diarrhea and nocturia, regulate the function of the stomach and spleen, and restore health from various ailments and debilitation caused by aging [13]. Meanwhile, numerous studies have shown that *Sipunculus nudus* and its extracts possess multiple activities, such as anti-radiation [14], anti-hypoxia [15], and anti-tumor [16]; promoting wound healing [17]; and immune regulation [18]. However, little attention has been paid to the potential benefits of applying *Sipunculus nudus* for cisplatin-induced kidney injury.

Therefore, in this study, one ultrafiltration fraction derived from enzymatic hydrolysis of *Sipunculus nudus* was obtained, and its characteristics were determined. This specific fraction was chosen due to its enriched content of bioactive peptides. This ultrafiltration fraction, named UFSH, has previously demonstrated potential protective effects against cisplatin-induced damage in HK-2 human renal tubular epithelial cells, making it a promising candidate for mitigating cisplatin-induced nephrotoxicity. Subsequently, the protective effects of UFSH were assessed in a cisplatin-induced nephrotoxicity mouse model using histological staining methods and by measuring the levels blood urea nitrogen (BUN) and serum creatinine (SCr). The potential mechanism underlying its protective effect was investigated by measuring the expression of inflammatory factors (IL-1β, IL-6, TNF-α) and the inflammatory mediator COX-2 in renal tissue, as well as the activities of antioxidant enzymes (SOD, GSH-Px) and the level of malondialdehyde (MDA). Finally, the protein expression of the MAPK/NF-κB signaling pathway were examined by Western blotting. This study may provide a strong foundation for the application of UFSH as therapeutic products and functional foods for cisplatin-induced nephrotoxicity.

## 2. Results

### 2.1. Amino Acid Composition of UFSH

Table 1 shows the amino acid composition of UFSH. Among 17 amino acids in UFSH, arginine (Arg, 3.94 g/100 g) was most the common, followed by alanine (Ala, 3.22 g/100 g), leucine (Leu, 3.22 g/100 g), aspartic acid (Asp, 2.08 g/100 g), glycine (Gly, 1.96 g/100 g), valine (Val, 1.80 g/100 g), and glutamic acid (Glu, 1.71 g/100 g). It is important to note that Glu and Asp have been reported to show strong antioxidant activity [19].

### 2.2. Molecular Weight Distribution

The molecular weight distribution of UFSH was determined by high-performance volume exclusion chromatography. As shown in Figure 1, UFSH contains macromolecular substances, small-molecule peptides, and amino acids. Macromolecular substances in UFSH with a molecular weight greater than 2500 Da account for 26.10% of the total compounds. Substances in UFSH with a molecular weight (MW) between 500 Da and 2500 Da account for 23.84%, and those with a molecular weight less than 500 Da account for 50.05%. Thus, the average molecular weight of UFSH is less than 2500 Da.

### 2.3. Peptide Sequence Identification of UFSH

All peptides in UFSH were identified by LC-MS/MS. In total, 625 peptides were identified in UFSH, and the molecular weight ranged from 640 to 2942 Da (Figure 2A). Further analysis of 625 peptides demonstrated that the molecular weights of peptides in UFSH were mostly below 2000 Da, accounting for 99.41% of the total peptides. The 12 peptides with the highest content in UFSH are shown in Table 2. It is noteworthy that among the 12 characteristic peptides, there are amino acids with higher antioxidant activity (such as Arg, Leu, and Pro) that exhibit strong reducibility. In UFSH, Phe-Ile-Ile-Asp-Lys-Gly-Asn-Leu-Arg (FIIDDKGNLR), Phe-Arg-Cys-Pro-Glu-Ala-Leu (FRCPEAL), Val-Gly-Gly-Leu-Asp-Glu-Arg-Arg (VGGLDERR), Ile-Ile-Ala-Pro-Glu-Ala-Pro-Arg (IIAPPER), and Glu-Tyr-Asp-Glu-Ser-Gly-Pro-Ser-Ile-Val (EYDESGPSIV) were the most abundant peptides (Figure 2B–F).

### 2.4. UFSH Attenuated Cisplatin-Induced Renal Dysfunction in Mice

In this study, a mouse model of chronic cisplatin-induced kidney injury was established to investigate whether UFSH alleviates cisplatin-induced nephrotoxicity. According to the experimental design, cisplatin (10 mg·kg^−1^) was injected intraperitoneally once a week to induce kidney injury in mice. The treatment group received daily oral administration of UFSH at different doses. Blood urea nitrogen (BUN) and serum creatinine (SCr) are two important indicators of kidney injury. As shown in Figure 3A,B, compared to the control group, cisplatin injection significantly elevated the serum CRE and BUN levels by 2.75- and 3.62-fold, respectively (*p* < 0.01), indicating the development of nephrotoxicity in the cisplatin-treated mice. Treatment with UFSH at medium and high doses significantly reduced the SCr level. Similarly, UFSH treatment at all three different doses reversed the increase in BUN in cisplatin-treated mice (*p* < 0.001 for each dose).

### 2.5. UFSH Attenuated Renal Histopathological Changes in Mice

To examine the effect of UFSH on cisplatin-induced renal histopathological damage, mouse kidney sections were stained with H&E and scored. As shown in Figure 4, the renal tissues in the control group exhibited completely normal renal structures, characterized by clear tubular and glomerular structures with clear and normal nuclei. In contrast, the kidney of the cisplatin-treated mice showed serious dilation of renal tubules, formation of casts and necrosis of tubular cells, indicating evident renal injury. However, treatment with UFSH markedly ameliorated histopathological damage in the kidney tissues, especially at the medium dose of 0.5 g/kg (Figure 4). However, treatment with UFSH significantly ameliorated histopathological damage in the kidney tissues, particularly at the medium dose of 0.5 g/kg (Figure 4).

### 2.6. UFSH Attenuated Renal Fibrosis Changes in Mice

Previous studies have shown that multiple cisplatin treatments can cause renal fibrosis [20]. The effect of UFSH on cisplatin-induced renal tubulointerstitial fibrosis was assessed using Masson’s trichrome staining. As shown in Figure 5 only a small amount of collagen fiber-positive area was observed in the tubular interstitium of the control group mice. However, after cisplatin injection, the stained collagen fibers in the renal tubular interstitial area were significantly increased. UFSH treatment significantly ameliorated renal fibrosis in the kidney tissues, especially at the lower and middle doses (*p* < 0.05) (Figure 5). These results indicate that UFSH treatment has an ameliorative effect on renal lesions and fibrosis in cisplatin-induced mice.

### 2.7. UFSH Inhibited the Expression of Renal KIM-1 in Renal

Kidney injury molecule-1 (KIM-1) is an important biomarker of renal tubular injury [21]. The gene expression of KIM-1 in the renal tissues was measured by qPCR. As indicated in Figure 6C, cisplatin exposure significantly increased the gene expression of KIM-1 in the kidney when compared to the control group. Treatment with UFSH significantly inhibited KIM-1 gene expression in a dose-dependent manner. To further confirm the qPCR results, the protein expression of KIM-1 in the renal samples was assessed using immunohistochemical staining. (Figure 6B). Cisplatin exposure increased the protein expression of KIM-1 compared to the control group. The elevated Kim-1 levels could be ameliorated by UFSH treatment. These results indicate that UFSH treatment improved renal tubular injury in mice.

### 2.8. UFSH Attenuated Cisplatin-Induced Inflammation

In order to investigate the effect of UFSH on cisplatin-induced inflammation, RT-PCR was first used to detect the mRNA expression of pro-inflammatory factors TNF-α and L-1β and the inflammatory mediator COX-2 in the kidney. Compared with the control group, the expressions of TNF-α, L-1β and COX-2 in the cisplatin-treated group were significantly increased (*p* < 0.001), and these increases were significantly reduced by UFSH treatment in a dose-dependent manner (*p* < 0.05) (Figure 7A–C). Concurrently, the protein expressions of pro-inflammatory cytokines TNF-α, IL-1β and IL-6 in the mouse kidney tissue were detected using ELISA kits. The protein expressions of TNF-α, IL-1β and IL-6 were significantly increased after cisplatin induction (*p* < 0.001). The treatment of UFSH significantly inhibited the protein expression of TNF-α at three doses (*p* < 0.01) (Figure 7D). At the same time, the protein expressions of IL-1β and IL-6 were significantly decreased by UFSH at low and medium doses (*p* < 0.05) (Figure 7E,F).

### 2.9. UFSH Reduced Cisplatin-Induced Oxidative Stress

In order to evaluate the effect of UFSH on cisplatin induced oxidative stress, the homogenate of mouse kidney tissue was extracted for the detection of the content of MDA and the activities SOD, GSH-PX. The content of MDA was increased, whereas the activities of SOD and GSH-PX were significantly decreased after cisplatin administration. Increased MDA was significantly inhibited by the treatment of UFSH at the middle dose (*p* < 0.05) (Figure 8A). Meanwhile, decreased activities of SOD and GSH-PX were significantly promoted by UFSH at the high dose and middle dose (*p* < 0.05) (Figure 8B).

### 2.10. Effects of UFSH on MAPK and NF-κB Pathways in Cisplatin-Induced Renal Injury

To further investigate the protective mechanism of UFSH in the cisplatin-treated mice model, we examined the effect of UFSH on mitogen-activated protein kinases (MAPK) and nuclear factor-κB (NF-κB) signaling pathways using Western blotting. As shown in Figure 9A, extracellular signal-regulated kinases (ERKs) and MAPK signaling proteins of p38 were activated after the repeated injection of cisplatin, and the upregulation of p-p38 and p-ERK phosphorylation was significantly inhibited by UFSH treatment at medium and higher doses (Figure 9A,C,D). The NF-κB signaling pathway is a classic inflammatory pathway, and it promotes the release of inflammatory factors in the cisplatin-induced renal injury model of mice. As shown in Figure 9B,E, cisplatin exposure promoted the phosphorylation level of NF-κB-p65 in the kidney of mice (*p* < 0.05). However, the phosphorylation level of p65 was significantly inhibited by UFSH at medium and high doses (*p* < 0.05).

### 2.11. UFSH Attenuated Cisplatin-Induced Apoptosis

As shown in Figure 9A, the protein expression of Bax was significantly increased, while the protein expression of Bcl-2 was significantly decreased by cisplatin administration. Bax protein expression was decreased, while Bcl-2 protein expression was increased after the addition of UFSH (Figure 10A). The ratio of Bax to Bcl-2 expression is an important indicator of apoptosis. As indicated in Figure 10B, the Bax/Bcl-2 ratio was significantly increased in the cisplatin-exposed group (*p* < 0.001), which could be decreased by UFSH treatment at the medium dose (*p* < 0.05) (Figure 10B).

## 3. Discussion

The clinical use of cisplatin has been limited due to its toxicity and several side effects, particularly nephrotoxicity. According to previous studies, the main mechanisms of cisplatin-induced chronic nephrotoxicity are primarily involved in oxidative stress, inflammation, and apoptosis. Previously, natural products [10,22,23,24] with anti-inflammatory and antioxidant properties have showed protective effects against cisplatin-induced nephrotoxicity. For example, hesperetin, a naturally occurring compound with abundant sources, significantly attenuated cisplatin-induced nephrotoxicity by inhibiting oxidative stress, inflammation, and apoptosis [22]. In addition, xanthohumol, a natural prenylated flavonoid extracted from the hop plant (*Humulus lupulus* L.), also showed a significant protective effect against cisplatin-induced nephrotoxicity by ameliorating inflammatory and oxidative responses [10]. Therefore, the anti-inflammatory regulation of cisplatin-induced oxidative stresses may provide new strategies for the future prevention and treatment of cisplatin-induced nephrotoxicity. In this study, we aimed to investigate the protective effect of an ultrafiltration fraction of *Sipunculus nudus* hydrolysate (UFSH) on cisplatin-induced nephrotoxicity in a low-dose repeated injection cisplatin mouse model.

Compared with a single injection of a large dose of cisplatin, the method of low-dose repeated injections of cisplatin is more in line with the dosage and course recommended by clinical cancer chemotherapy guidelines, which can simulate the long-term chronic renal toxicity and side effects caused by cisplatin [25]. The pathophysiology of cisplatin-induced nephrotoxicity involves several stages [26]. Renal tubular damage in the kidney is the most common characteristic phenomena [8]. Urea and creatinine, which are considered waste products of the body, are usually filtered by the kidney and excreted in urine [27]. However, when kidney function deteriorates, they gradually accumulate in the body and can be easily detected in the serum [28]. Therefore, BUN and SCr levels are considered to be two important indicators for diagnosing chronic renal injury. In line with previous studies [25], BUN and SCr levels were significantly elevated in mice subjected to low-dose repeated injections of cisplatin. In the current study, we demonstrated that the administration of UFSH improved renal function injury. In the cisplatin-induced CKD model described here, loss of brush border, tubular cell necrosis, and cast formation were detected, as previously reported. In addition, an increase in the expression of Kim-1, a marker of kidney injury, was also observed in the model group through immunohistochemical staining. However, the administration of UFSH significantly improved these changes, indicating that UFSH protects against the decline in kidney function in cisplatin-induced CKD.

Until now, the mechanism of cisplatin-induced nephrotoxicity has not been fully clarified. However, according to previous studies, oxidative stress and inflammation are important factors involved in cisplatin-induced nephrotoxicity [7]. The kidney is a highly metabolic organ. Mitochondria are rich in oxidation reactions, which makes it vulnerable to the damage caused by oxidative stress and induce the production of free radical such as reactive oxygen species (ROS) [29]. Many antioxidant enzymes (SOD, GSH-Px, etc.) exist in renal cells to prevent cell or tissue damage by scavenging ROS [30]. The level of malondialdehyde (MDA) is an indicator of lipid peroxidation and is often referred to as a sign of oxidative stress and increased free radicals [31]. The current study showed that UFSH significantly promoted SOD activity in a dose-dependent manner. At the same time, UFSH can also increase the activity of GSH-Px, whereas it reduced the production of MDA. These findings strengthen the hypothesis that the renal protective effect of UFSH can be attributed to its free radical scavenging and antioxidant properties.

Cisplatin-induced nephrotoxicity is also closely related to inflammation, and persistent low-grade chronic inflammation has been recognized as an important part of CKD [32]. The expression of various inflammatory factors and inflammatory mediators can be significantly increased under cisplatin induction [4]. According to previous studies, the elevation of pro-inflammatory cytokine levels is mediated by the activation of the signaling cascade, including MAPKs and NF-κB. Mitogen-activated protein kinases (MAPKs) are members of intracellular protein kinases, including ERK, JNK, and p38 [33], which have the function of regulating a variety of cellular processes. Hyperphosphorylation of the MAPK molecules ultimately activate the transcription factor NF-κB and subsequent production of inflammatory molecules [34]. In line with previous studies [35], the levels of pro-inflammatory cytokines (IL-1β, IL-6, and TNF-α) were elevated in cisplatin-injected mice. However, UFSH significantly inhibited the mRNA expression levels of TNF-α, IL-6 and COX-2, as well as the secretion of TNF-α, IL-1β, and IL-6 in cisplatin-exposed mice. Moreover, UFSH not only reduced the phosphorylation of p38 and ERK proteins in the MAPK signaling pathway but also inhibited the phosphorylation of NF-κB p65. These findings suggest that UFSH plays a protective role in the kidney through its anti-inflammatory activity by deactivating the MAPK and NF-κB pathways.

Except oxidative stress and inflammatory response, apoptosis also plays an important role in cisplatin-induced nephrotoxicity, and renal tubular cell damage has been recognized as an important part of CKD. Cisplatin-induced renal epithelial cell death is associated with several apoptotic pathways [36,37]. Numerous studies have shown that the Bcl-2 family plays an extremely important regulatory role in the apoptotic pathway. Under normal conditions, Bcl-2 can maintain the integrity of the mitochondrial outer membrane by forming heterodimers with Bax, preventing the release of cytochrome c from mitochondria to the cytoplasm, and inhibiting mitochondrial apoptosis [38]. The overexpression of Bax in cells can enhance the activation effect of intracellular Caspase and antagonize the protective effect of Bcl-2 on cells, thus causing cell apoptosis [39]. By detecting the apoptotic signaling pathway, UFSH can reduce the mRNA level and protein expression of Bax and decrease the Bax/Bcl-2 ratio, which indicates that UFSH has a certain anti-apoptotic effect.

In conclusion, our results demonstrate that UFSH attenuates cisplatin-induced nephrotoxicity in mice by suppressing oxidative stress, inflammatory response, and apoptosis. This study shows that UFSH exhibits prominent kidney-protective activity due to its antioxidant and anti-inflammatory properties, as well as by regulating the abnormal expression of KIM-1. A schematic explanation for cisplatin-induced nephrotoxicity and the role of UFSH is shown in Figure 11. This study suggests that UFSH may serve as therapeutic products or functional foods for the treatment of nephrotoxicity in patients receiving cisplatin therapy.

## 4. Materials and Methods

### 4.1. Preparation of Ultrafiltration Components Extracted by Enzymatic Hydrolysis of Sipunculus nudus

Fresh *Sipunculus nudus* (SN) specimens were obtained from the sea adjacent to Zhanjiang city in China. After removing visible sand, SN meat (around 10 kg for this study) was first ground with a meat mincer and then homogenized with 100 L of pure water in a hydrolysis tank. Subsequently, the homogenate was hydrolyzed at 45 °C using trypsin (4 × 10^4^ U/g protein, pH = 7.5) for 4 h. During the enzymatic hydrolysis, attention was paid to monitoring the pH value. After enzymatic hydrolysis, the enzyme in the solution was denatured by heating treatment (90 °C, 30 min). Following the inactivation steps, the enzymatic hydrolysate was centrifuged at 12,000 rpm for 20 min. Finally, the supernatants were gradually ultrafiltrated using 50,000 Da and 2500 Da ultrafiltration membranes (Millipore, Darmstadt, Germany). Three ultrafiltration fractions were collected. Among them, the first ultrafiltration fraction retained by the 50,000 Da ultrafiltration membrane (named UFSH in this study) showed the best effect on improving cisplatin-induced renal tubular epithelial cell injury and was selected for further assessment in the animal experiment.

### 4.2. Amino Acid Composition Analysis

An amino acid standard mixed solution was initially prepared as follows: A mixed standard amino acid reserve solution containing 17 kinds of amino acids was diluted to various concentrations using a 0.01 mmol/L hydrochloric acid solution. Then, 0.1 g of UFSH was added to a 10 mL 6 mol/L hydrochloric acid solution in a hydrolysis tube. The tube was subsequently hydrolyzed for 24 h in an electric blast incubator at 110 °C. After hydrolysis, the cap of the tube was unscrewed, and the hydrochloric acid was evaporated at 80 °C. The solution was then brought to an appropriate volume, and approximately 1 mL was taken for analysis. Standard curves were constructed using the peak times of different concentrations of standard amino acids, which were then used to calculate the content of each amino acid in the sample. The chromatographic conditions were as follows: column, Thermo AminoPac PA-10 (4 mm × 250 mm, 10 µm); flow rate, 0.24 mL/min; sample size, 25 µL; column temperature, 30 °C. The chromatogram was recorded, and the retention time and peak area of each amino acid were determined. Finally, the peak area of the internal standard and each amino acid was measured, and the content of each amino acid was obtained.

### 4.3. Determination of Molecular Weight Distribution

The molecular weight distribution of UFSH was determined by high-performance size exclusion chromatography (GE Superdex 30 Increase 10/300 GL, Boston, MA, USA). A 25 μL sample was loaded onto a Protein-Pak 60 column (Waters Corp., Milford, MA, USA) for HPLC elution with a 0.5 mL/min mobile phase (0.05 mol/L PBS buffer, pH 7.2) and detected at a 220 nm wavelength. To determine the molecular weight distribution of UFSH, four kinds of proteins including cytochrome C, α-lactalbumin, aprotinin, insulin-B, and bacitracin were used as standard molecular weight markers during HPLC elution. The regression equation between the logarithm of molecular weight M and the elution retention time t was established. The molecular weight distribution of the UFSH was studied using this standard curve regression equation: log(M) = −0.2315t + 6.3412, yielding R ^2^ = 0.9949.

### 4.4. Identification of Peptides in UFSH Using LC-MS/MS

Peptides in UFSH were analyzed by liquid mass spectrometry (LC-MS/MS) according to a previous study [40], and the identification results were obtained after matching data with PEAKS Studio10.6 software by Biotech-Pack Scientific Co., Ltd. (Beijing, China).

### 4.5. Animal Models and Treatment

Adult male C57BL/6 J mice (7–8 weeks old, weighing 20–25 g) were sourced from Spife Biotechnology (Beijing, China) and housed under pathogen-free conditions in a climate-controlled room (temperature 19–21 °C, humidity 50–60%). Throughout the experiment, all mice had free access to food and water. Cisplatin (D8810; Solarbio Biotechnology Co., Ltd., Beijing, China) was prepared in 0.9% normal saline, and UFSH was dissolved in 0.05% carboxymethyl cellulose (CMC). Before the experimental procedure began, the mice were acclimated to the laboratory environment for at least 1 week. Subsequently, 40 adult male C57BL/6 J mice were randomly assigned to 5 groups (n = 8 per group): control group (CON), model group (CIS), cisplatin + UFSH low-dose group (100 mg/kg.bw, CIS+UFSH-L), cisplatin + UFSH medium-dose group (500 mg/kg.bw, CIS+UFSH-M), and cisplatin + UFSH high-dose group (1000 mg/kg.bw, CIS+UFSH-H).

Mice in the model and UFSH-treated groups received intraperitoneal injections of cisplatin weekly (10 mg/kg for the first two injections, and 7 mg/kg for the last two injections), while the control group received saline injections. In addition to the weekly cisplatin injections, mice in the treatment groups were administered UFSH orally at different doses daily for four weeks. Mice in the control and model groups were treated with saline solution instead. This study was approved by the Animal Care and Use Committee of Guangdong Medical University (2 March 2022; GDY2202193), and experiments were conducted in a blinded manner.

At the end of the experiment, mice were euthanized. Blood was collected, and serum was separated and stored at −80 °C for future use. Kidneys were also collected at the end of the study. Each kidney sample was divided into two parts: one part (0.2 g) was fixed in paraformaldehyde for at least 24 h, dehydrated in a graded ethanol series, and embedded in paraffin for histopathological analysis; the second part was snap-frozen in liquid nitrogen and stored at −80 °C for subsequent ELISA, RT-qPCR, and Western blotting analyses.

### 4.6. Renal Function Test

Serum samples were obtained through centrifugation at 1500 rpm for 20 min. The levels of serum creatinine (SCr) and blood urea nitrogen (BUN) were measured following the instructions provided in the manufacturer’s kit (Nanjing Jiancheng Bioengineering Institute, Nanjing, China).

### 4.7. Histological Examination

H&E staining: Paraffin-embedded renal tissue sections were dewaxed to water, and the sections were stained with an H&E staining kit (G1003) according to the manufacturer’s kit instruction (Servicebio Biotechnology Co., Ltd., Wuhan, China). The severity of renal injury was evaluated by optical microscope with 200× magnification, and the lesions were scored according to [41].

Masson trichrome staining: The sections were dewaxed to water then stained with the Masson trichrome staining kit. The staining kit was purchased from Servicebio Biotechnology Co., Ltd. (Wuhan, China). The staining method was conducted in accordance with their respective staining procedures with no modification. Each stained sample section was evaluated and quantified using the ImageJ Version 1.54p software program.

### 4.8. Immunohistochemical Stainning

The expression of kidney injury molecule-1 (KIM-1) was assessed via immunohistochemical staining. Paraffin sections were dewaxed and subjected to antigen retrieval in a citric acid buffer (pH 6.0) using a microwave oven. After cooling, sections were washed in PBS (pH 7.4), treated with hydrogen peroxide to inactivate endogenous peroxidase, and blocked with BSA. They were then incubated with a mouse polyclonal anti-KIM-1 antibody (AF1817; R&D, Minneapolis, MN, USA) overnight, followed by an HRP-labeled secondary antibody. DAB was used for chromogen development, controlled under a microscope. Sections were counterstained with hematoxylin, dehydrated, and sealed. Images of at least 10 fields (200×) were captured for analysis.

### 4.9. Enzyme-Linked Immunosorbent Assay

Kidney tissues (0.2 g) were homogenized at 10,000 rpm for 20 s in nine portions of PBS solution (1.8 mL), followed by centrifugation at 12,000 rpm for 5 min. The supernatant was collected, and the levels of IL-6, IL-1β, and TNF-α were measured using ELISA kits according to the manufacturer’s instructions (Jiangsu Meimian industrial Co., Ltd., YanCheng, China).

### 4.10. Determination of SOD, GSH-Px and MDA in Renal Tissue

Following the manufacturer’s instructions, the activities of SOD and GSH-Px in renal tissue were measured using commercial kits from Nanjing Jiancheng Bio-engineering Institute (Nanjing, China). Additionally, the concentration of MDA in renal tissue was determined using a commercial MDA detection kit from Beyotime Biotechnology (Shanghai, China).

### 4.11. Western Blotting

Kidney tissue samples were homogenized in a protease inhibitor-containing lysate and centrifuged to collect the supernatant, whose protein concentration was measured using a BCA kit (Beyotime Biotechnology, Shanghai, China). Then, 30 μg of protein was run on 10% SDS-PAGE gels and transferred to PVDF membranes. After blocking with 5% BSA, the membranes were incubated with primary antibodies overnight and secondary antibodies for 2 h. Bands were detected using BeyoECL Moon and analyzed with a chemiluminescence system (Tanon 5200, Shanghai, China).

The primary antibodies used in this study included NF-κB p65 (sc-8008, diluted 1:1000) and p-NF-κB p65 (sc-136548, diluted 1:1000), which were purchased from Santa Cruz Biotechnology, Dallas, TX, USA. Other primary antibodies, such as Bax (GB12690, diluted 1:1000), Bcl-2 (GB48675, diluted 1:700), and the secondary antibody for GAPDH (GB12002, diluted 1:1000), were obtained from Servicebio Biotechnology Co., Ltd., Wuhan, China.

### 4.12. Real-Time PCR

A rapid RNA extraction reagent (AG21101; Accurate biology, Changsha, China) was used to extract total RNA from mouse kidney tissues. Then, 1 μg of total RNA was used to generate cDNA using a cDNA synthesis reagent (K1622; Thermo Fisher, Waltham, MA, USA). Using TB Green^®^ Premix Ex Taq^TM^ II (RR820B; TAKARA BIO, Kusatsu, Japan) in qTOWER3 real-time fluorescent quantitative PCR (analytikjena, Jena, Germany) was performed using the amplification primers in Table 3. The PCR reactions were prepared from 1 mL of cDNA mixed with 2× Quantitect SYBR Green PCR Master Mix (Qiagen), 0.5 μM forward and reverse primers (Table 3), and 1 μL of RNase-free water. The reactions were carried out on a Corbett Life Science (Sydney, Australia) Rotor-Gene 6000 system using the following cycling program: 94 °C for 2 min (1 cycle); 94 °C for 30 s, 51 °C, 53 °C or 56 °C for different genes for 30 s, 72 °C for 30 s (35 cycles); and 72 °C for 7 min (1 cycle). Gene expression levels were normalized to the mRNA expression of the housekeeping gene GAPDH (relative quantification) with the ΔΔCT correction.

### 4.13. Statistical Analysis

All data are presented as the mean ± standard error (SEM) and were analyzed by SPSS software version 17. Experimental values were evaluated by one-way analysis of variation after following the LSD method as multiple comparisons between groups. Differences were considered significant if the *p*-value was less than 0.05.

## 5. Conclusions

In summary, the present study demonstrated for the first time that UFSH significantly attenuates cisplatin-induced nephrotoxicity. The effects of UFSH may work via reducing inflammation and oxidation and mitigating apoptosis, as well as regulating MAPK/NF-κB signaling. The antioxidant and anti-inflammatory effects in this study are possibly due to the activities of one or more or mixtures of these identified classes of compounds.

## Figures and Tables

**Figure 1 marinedrugs-23-00100-f001:**
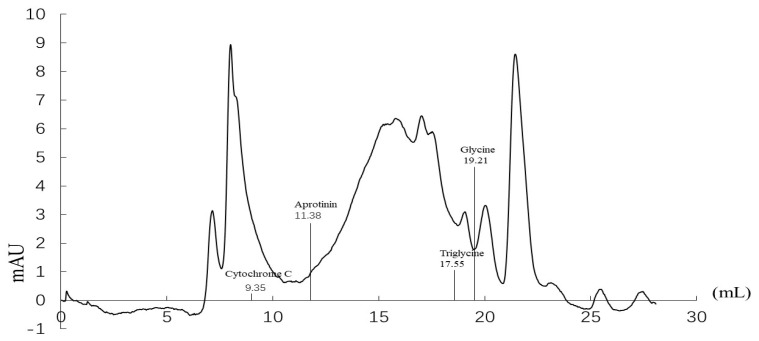
Distribution of the molecular weight and mass spectrometry of UFSH.

**Figure 2 marinedrugs-23-00100-f002:**
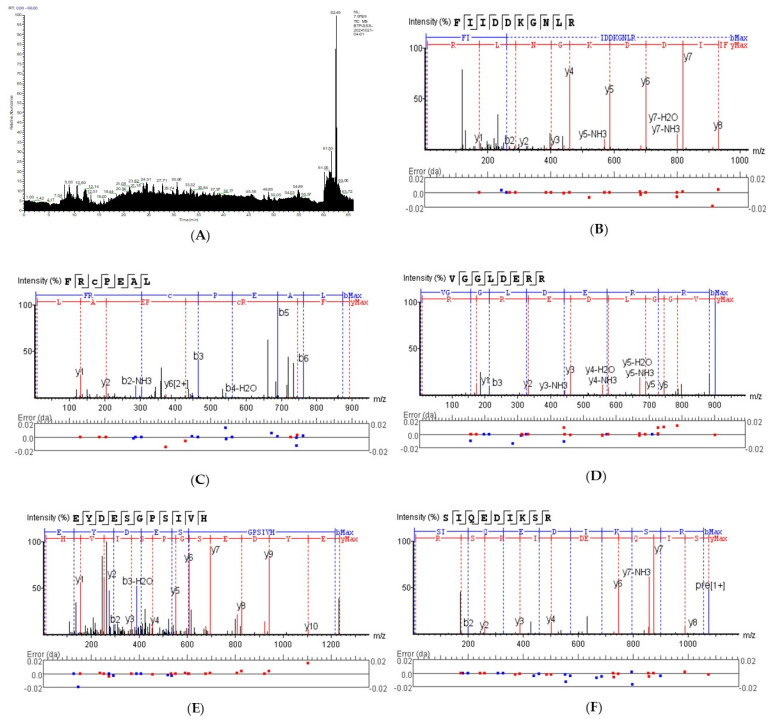
Peptide sequence identification of UFSH. (**A**) Total ion chromatogram of UFSH; (**B**) secondary mass spectrum of peptide FIIDDKGNLR in UFSH; (**C**) secondary mass spectrum of peptide FRCPEAL in UFSH; (**D**) secondary mass spectrum of peptide VGGLDERR in UFSH; (**E**) secondary mass spectrum of peptide IIAPPER in UFSH; (**F**) secondary mass spectrum of peptide EYDESGPSIV in UFSH.

**Figure 3 marinedrugs-23-00100-f003:**
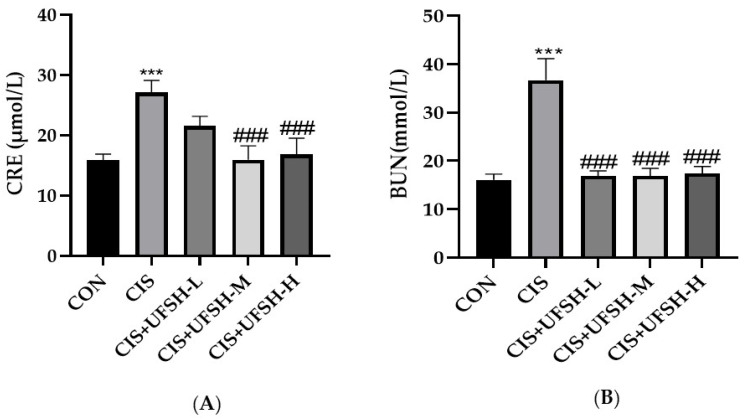
Effects of UFSH on cisplatin-induced renal function injury in mice. (**A**) Serum creatinine level of mice with different treatment. (**B**) Serum urea nitrogen level of mice with different treatments. *** *p* < 0.001, compared with the control group, ^###^
*p* < 0.001 compared with the cisplatin-injected group.

**Figure 4 marinedrugs-23-00100-f004:**
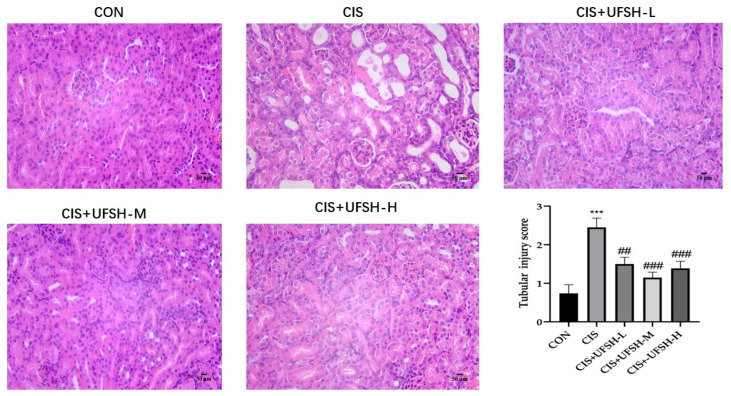
The effect of UFSH on renal histopathology changes in cisplatin-induced mice. Renal tubular injury scores were analyzed by H&E staining (scale bars: 50 μm, magnification 400×). *** *p* < 0.001, compared with the control group. ^##^
*p* < 0.01 compared with the cisplatin-injected group. ^###^
*p* < 0.001 compared with the cisplatin-injected group.

**Figure 5 marinedrugs-23-00100-f005:**
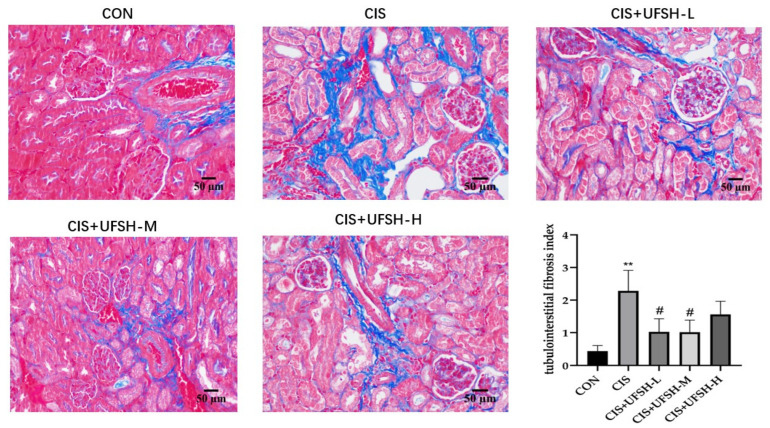
The effect of UFSH on cisplatin-induced renal fibrosis changes. Relative renal fibrosis index analyzed by Masson’s trichrome staining; slices stained with dark blue indicate collagen fiber, and red indicates cytoplasm and muscle fiber. ** *p* < 0.01, compared with the control group. # *p* < 0.05, compared with the cisplatin-injected group. n = 8 mice in each group.

**Figure 6 marinedrugs-23-00100-f006:**
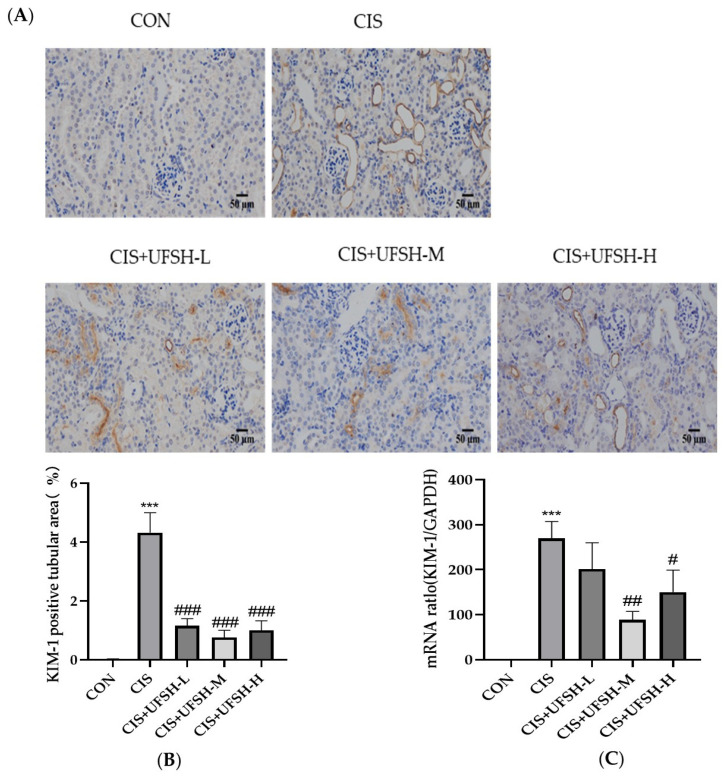
The effect of UFSH on the gene and protein expressions of KIM-1 in the kidneys of mice. (**A**) Representative photographs of renal Kim-1 expression by IHC (scale bars: 50 μm, magnification 400×). (**B**) Relative KIM-1 positive expression in the kidney was assessed by comparing the number of KIM-1 positive cells in different groups vs. the control group from 10 equal sections of immune-stained kidney tissue per animal. (**C**) The effect of UFSH treatment on KIM-1 mRNA expression in the kidney tissue of mice. *** *p* < 0.001, versus control group. ### *p* < 0.001, ## *p* < 0.01, # *p* < 0.05, versus cisplatin-injected group. *n* = 8 mice in each group.

**Figure 7 marinedrugs-23-00100-f007:**
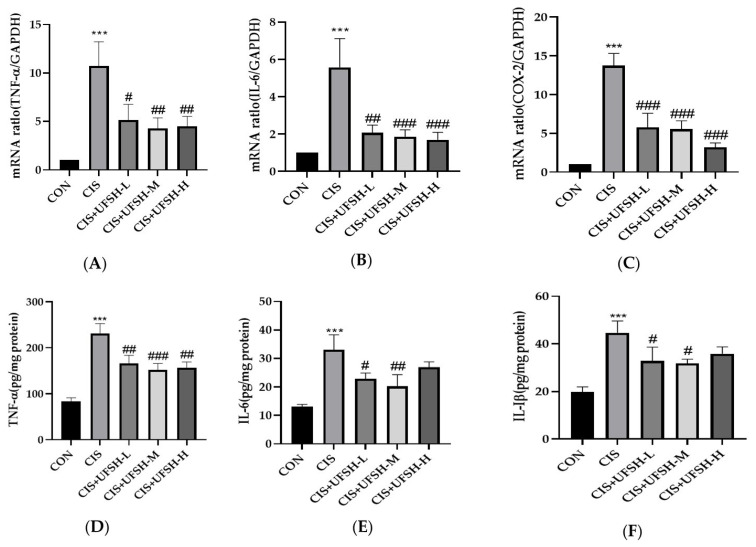
The effect of UFSH on cisplatin-induced inflammation in mouse kidney tissue. (**A**) The relative mRNA expressions of TNF-α in different groups. (**B**) The relative mRNA expressions of IL-6 in different groups. (**C**) The relative mRNA expressions of COX-2 in different groups. (**D**) The protein expression level of TNF-α in mice kidney tissues; (**E**) the protein expression level of IL-6 in mice kidney tissues; (**F**) the protein expression level of IL-1β in mice kidney tissues. *** *p* < 0.001, compared with the control group. # *p* < 0.05, ## *p* < 0.01, ### *p* < 0.001 were compared with the cisplatin-injected group. *n* = 8 mice in each group.

**Figure 8 marinedrugs-23-00100-f008:**
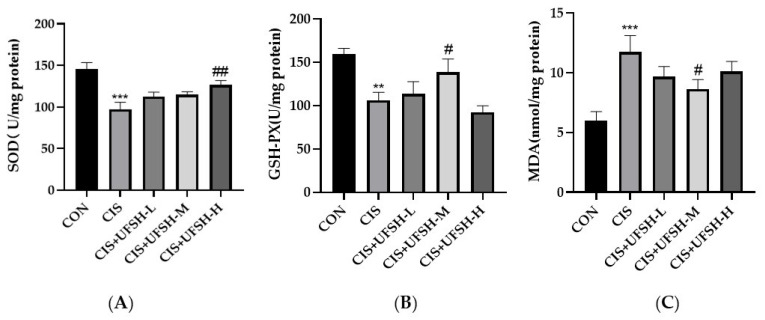
The effects of UFSH on cisplatin-induced oxidative stress. (**A**) The activities of renal SOD in different groups; (**B**) the activities of renal GSH-PX in different groups; (**C**) the levels of renal malondialdehyde (MDA) in different groups; each bar represents the mean ± SEM. *** *p* < 0.001, ** *p* < 0.01 compared with the control group. # *p* < 0.05, ## *p* < 0.01 compared with the cisplatin group. *n* = 8 mice in each group.

**Figure 9 marinedrugs-23-00100-f009:**
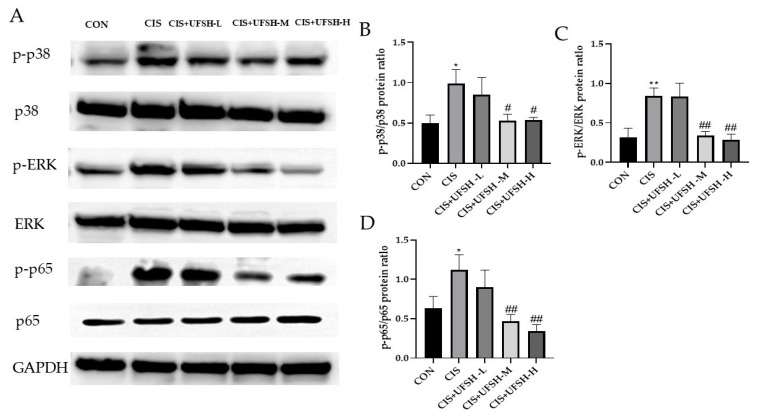
The effects of UFSH on MAPK and NF-κB pathways in the kidneys of cisplatin-induced mice. (**A**) Representative images of MAPK (p38, ERK and p-p38, p-ERK) and p-p65 measured by Western blotting. (**B**) Representative images of NF-κB (p-p65) measured by Western blotting. (**C**) Quantitative analysis of p-p38/p38 and p-ERK/ERK bands. (**D**) Quantitative analysis of p-p65 expression. The data were normalized to the protein expression of the housekeeping gene GAPDH using the delta-Ct method and expressed as the mean ± S.E.M. fold change relative to the control. * *p* < 0.05, ** *p* < 0.01 compared with the control group. # *p* < 0.05, ## *p* < 0.01 compared with the cisplatin group.

**Figure 10 marinedrugs-23-00100-f010:**
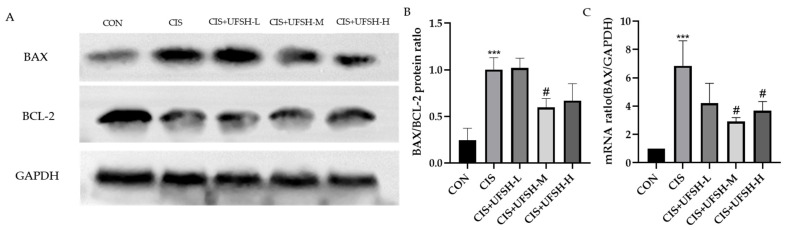
The effects of UFSH on cisplatin-induced apoptosis measured by Western blotting. (**A**) The protein expressions of Bax and Bcl-2 in mouse kidney tissue. (**B**) Bax/Bcl-2 protein ratio in mouse kidney tissues. (**C**) Real-time PCR for BAX. Each bar represents the mean ± SEM. *** *p* < 0.001, compared with the blank control group. # *p* < 0.05, compared with the cisplatin group.

**Figure 11 marinedrugs-23-00100-f011:**
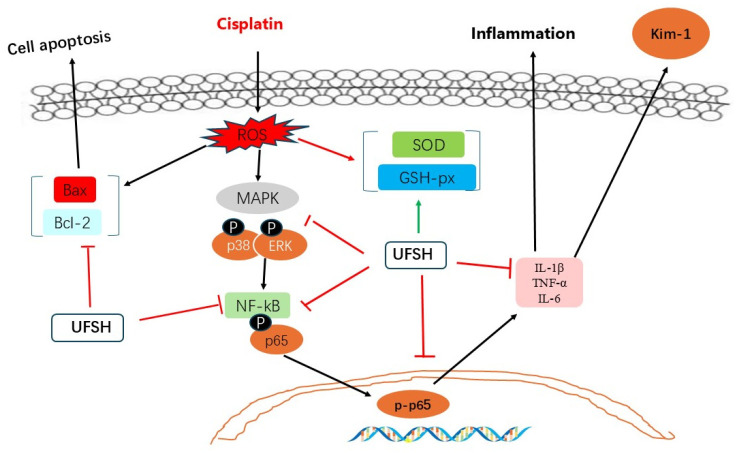
Possible mechanism by which UFSH attenuates cisplatin-induced nephrotoxicity.

**Table 1 marinedrugs-23-00100-t001:** Amino acid composition of UFSH.

Amino Acids	Hydrolyzed Amino Acids (g/100 g)	Amino Acids	Hydrolyzed Amino Acids (g/100 g)
Arginine (Arg)	3.94	Leucine (Leu) *	2.33
Lysine (Lys) *	1.15	Methionine (Met) *	0.58
Alanine (Ala)	3.22	Histidine (His)	0.56
Threonine (Thr) *	1.10	Phenylalanine (Phe)	1.21
Glycine (Gly)	1.96	Glutamic (Glu)	1.71
Valine (Val) *	1.80	Aspartic (Asp)	2.08
Serine + Proline (Ser + Pro)	0.90	Cysteine (Cys)	0.10
Isoleucine (Iso) *	1.67	Tyrosine (Tyr) *	1.17
Total	25.49		

Note: * essential amino acid.

**Table 2 marinedrugs-23-00100-t002:** Peptide sequence, length, mass, source, and content of the 12 peptides with the highest content in UFSH.

No.	Peptide Sequence	Length	Mass	Contents (%)
1	FIIDDKGNLR	10	1189.65	5.44
2	FRCPEAL	7	891.43	2.18
3	VGGLDERR	8	900.48	2.06
4	IIAPPER	7	794.47	1.90
5	EYDESGPSIV	11	1231.54	1.77
6	SIQEDIKSR	9	1074.57	1.51
7	RELPGHT	7	808.42	1.45
8	KCDVDIR	7	904.44	1.41
9	ITKPAPQ	7	753.44	1.38
10	RVAPEEHP	8	933.47	1.28
11	IIAPPERK	8	922.56	1.12
12	ILGPGGK	8	640.39	1.08

**Table 3 marinedrugs-23-00100-t003:** Mouse primer sequences of TNF-α, IL-6, COX-2, KIM-1, BAX and GAPDH.

Gene	Primary Sequence (5′→3′)
Mouse TNF-α	F: CATCTTCTCAAAATTCGAGTGACAA
	R: TGGGAGTAGACAAGGTACAACCC
Mouse IL-6	F: ACAAAGCCAGAGTCCTTCAGAGAG
	R: TTGGATGGTCTTGGTCCTTAGCCA
Mouse COX-2	F: AGGACTCTGCTCACGAAGGA R: TGACATGGATTGGAACAGCA
Mouse KIM-1	F: ACATATCGTGGAATCACAACGAC R: ACTGCTCTTCTGATAGGTGACA
Mouse BAX	F: TGGAGATGAACTGGACAGCAATAT R: GCAAAGTAGAAGAGGGCAACCAC
Mouse GAPDH	F: GTCTTCACTACCATGGAGAAGG R: TCATGGATGACCTTGGCCAG

## Data Availability

The data used to support the findings of this study are included within the article. Further inquiries can be directed to the corresponding author.

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
