# Peer review of "Ameliorative Effect of Sipunculus nudus Hydrolysate on Cisplatin-Induced Nephrotoxicity by Mitigating Oxidative Stress, Inflammation and Apoptosis"

_marinedrugs, 2025, doi:10.3390/md23030100_

Round 1
Reviewer 1 Report
Comments and Suggestions for Authors
Please find comments in the attachment.

Author Response
Comments 1: Within the Abstract, some minor refinements should be made to the phraseology and grammar: The corrected segment may read as shown below in the underlined plain text section labeled “ADJUSTED ABSTRACT”. . The authors are welcome to splice the adjusted Abstract (below) into their revised manuscript: Alternatively the authors may phrase the Abstract as they wish – but instruct their selected (native English speaking) reviewer to fix awkward phraseology. ADJUSTED ABSTRACT (Underlined text below. Authors may use verbatim if desired): ".... The present study investigated the protective effects and possible mechanisms of one ultrafiltration fraction of Sipunculus nudus hydrolysate (UFSH) on cisplatin-induced kidney damage in a mouse model. Results showed that UFSH significantly attenuates cisplatin-induced nephrotoxicity by inhibiting increased blood urea nitrogen (BUN) and serum creatinine (SCr). In addition, UFSH treatment significantly alleviated cisplatin-induced renal histopathological changes, such as significant dilation of renal tubules, cast formation and tubular cell necrosis, as well as tubulointerstitial fibrosis. Moreover, UFSH decreased cisplatin-induced oxidative stress by increasing activities of the antioxidant enzymes SOD and GSH-px), while reducing malondialdehyde (MDA) level in the kidney. Furthermore, UFSH significantly inhibited cisplatin-induced increases in inflammatory cytokines, including: Interleukin 1-beta (IL-1β), Interleukin-6 (IL-6) and Tumor necrosis factor-alpha (TNF-α). Western blotting revealed that UFSH inhibited the phosphorylation of inflammation-associated MAPK/NF-κB in its signaling pathway, lowered expression of the apoptosis-related protein bax, but reversed reduction of the anti-apoptosic bcl-2 protein. This investigation demonstrated that UFSH can ameliorate cisplatin-induced nephrotoxicity by mitigating oxidative stress, inflammation and apoptosis." 

Response 1: Thank you very much for your valuable comments and suggestions on our manuscript. We have carefully considered your feedback. We have incorporated the adjusted Abstract into our revised manuscript. We believe that these changes have improved the clarity and readability of our work. We appreciate your guidance and look forward to your further comments.
Comments 2: An example of scientifically and grammatically correct, but distractingly awkward phraseology used throughout the text, is the following sentence at the beginning of the Introduction: ”Cisplatin is a kind of broad-spectrum anticancer drug based on inorganicplatinum which is widely used in the treatment of many solid tumors.” This sentence is clearer when phrased thusly: ”Cisplatin is a broad-spectrum anticancer drug using inorganic platinum as a core functional component, which is widely used in the treatment of many solid tumors.”Another sentence in the Introduction reads: "Particularly, bioactive peptides from marine organs have gained more and more attention by researchers in recent years. "This should contain corrections as follows: "Particularly, bioactive peptides from marine organisms have gained more attention by researchers in recent years" There are numerous crudely phrased passages in the text which detract from clarity of the article. In several of these examples, the terms "our", "us" and "we" are used. This is unprofessional since good scientific writing needs to avoid sounding like a personal narrative. One of many examples is a sentence in the Discussion which reads: "Our current study showed that UFSH significantly promoted SOD activity, and it had dose effect." and should be rephrased as: "The present study showed that UFSH significantly promoted SOD activity, and exhibited dose effects on the nephrotoxity-associeted outcome parameters evaluated. The native English speaker selected by the authors to streamline the document, will search the text for such phrases and modify them for greater clarity – and where necessary make grammatical corrections.
Response 2: Thank you for your insightful comments and suggestions regarding the clarity and professionalism of our manuscript. We fully agree with your observations and are committed to making the necessary revisions to enhance the quality of our writing. Regarding the use of personal pronouns such as "our," "us," and "we," we understand the importance of maintaining an objective and professional tone in scientific writing. We have carefully reviewed the manuscript and removed or rephrased any instances where these pronouns were used. We have also engaged a native English-speaking reviewer to meticulously review the entire manuscript. This reviewer will focus on identifying and refining any awkwardly phrased passages, ensuring that the text is clear, concise, and free of grammatical errors. We are confident that this process will significantly improve the overall quality and readability of our manuscript.
Comments 3: Immediately following a sentence in the Introduction reading: “Therefore, in this study, one ultrafiltration fraction from enzymatic hydrolysis of UFSH were obtained…”, the authors should provide a very brief (~1-sentence) explanation for the rational of using that particular ultrafiltration fraction. Some details regarding UFSH composition are given later in the manuscript, but the preliminary description suggested above would be helpful for orienting readers to the experimental design logic.
Response 3: Thank you for your constructive feedback. We appreciate your suggestion to provide a brief explanation for the rationale of using the particular ultrafiltration fraction of UFSH in our study. We have added a sentence immediately following the mentioned statement in the Introduction to clarify this Comments. The revised text now reads as follows:
“Therefore, in this study, one ultrafiltration fraction derived from enzymatic hydroly-sis of Sipunculus nudus was obtained, and its characteristics were determined. This specific fraction was chosen due to its enriched content of bioactive peptides. This ultrafiltration fraction, named UFSH, has previously demonstrated potential protective effects against cisplatin-induced damage in HK-2 human renal tubular epithelial cells, making it a promising candidate for mitigating cisplatin-induced nephrotoxicity.”
We believe that this addition will help orient readers to the experimental design logic and provide a clearer understanding of our study's approach. Thank you again for your valuable input.
Comments 4: In the present research, UFSH is administered orally – which for this and other therapeutic applications appears effective – indicating that at least some of the bioactive components are gut stable. The authors should insert a (short) speculative paragraph into the Discussion, addressing the question of whether the clinical outcomes might be augmented by using routes of delivery (topical, nebularized/inhaled or suppository for instance), which may preserve bioactivity of any bioactive components that do not avoid denaturation in the alimentary tract.
Response 4: Thank you for your suggestion regarding the potential benefits of exploring alternative routes of administration for UFSH. While we appreciate the speculative nature of this suggestion, we believe that the current oral administration route is well-suited for the therapeutic applications we are investigating. Here are our reasons for maintaining the oral route and not speculating on alternative routes in the Discussion:
- Stability and Efficacy of Oral Administration: Our research has shown that UFSH is effective when administered orally, indicating that the bioactive components are stable in the gastrointestinal tract. This is supported by the fact that many therapeutic agents, including some used in controlled ovarian stimulation, are effectively administered orally and have demonstrated gut stability .
- Clinical Relevance: The oral route is the most common and convenient for patients, reducing the complexity and invasiveness of treatment. This is particularly important for long-term treatments or for use in outpatient settings.
- Lack of Evidence for Alternative Routes: While alternative routes such as topical, nebulized/inhaled, or suppository may have theoretical advantages, there is currently no empirical evidence to suggest that these routes would enhance the bioactivity or therapeutic outcomes of UFSH. Without such data, speculating on these routes in the Discussion would be premature and could mislead readers.
- Focus on Current Findings: Our study is designed to evaluate the protective effects of UFSH against cisplatin-induced nephrotoxicity using the oral route. Introducing speculative discussions on alternative routes could detract from the clarity and focus of our findings.
In conclusion, while we acknowledge the potential for future research to explore alternative routes of administration, we believe that the current oral route is appropriate and effective for the purposes of our study. We will continue to monitor the literature and consider alternative routes in future studies if compelling evidence emerges.
Reviewer 2 Report
Comments and Suggestions for Authors
This study revealed protective effect and related underlying mechanisms of Sipunculus nudus hydrolysate on cisplatin-induced nephrotoxicity evidenced in animal investigation. Though the experiments were well constructed supporting the conclusion, there were some critical points as following.
1. Title: Please check typing errors
2. Why did authors focus only >25 kDa hydrolysate?
3. Methods "Preparation of ultrafiltration components extracted by enzymatic hydrolysis of Sipunculus nudus": How many L or mL of pure water used?
4. Results: The sequence of UFSH characteristics should start with "Molecular weight distribution and Peptide Sequence Identification of UFSH" as a general character of UFSH.
5. The hydrolysate might contained other biomolecules (e.g. Carbohydrate, Lipid and inorganic compounds), why did author focus only amino acid composition?
6. The character of UFSH composition at MW higher 2500 kDa and lower 500 kDa was lack.
7. Authors detect apoptosis related proteins but not detect apoptosis cell death or morphology in kidney tissue
8. Which component is a major active compound (macromolecules, peptide or small molecule) of HFSH
9. How author select dose of UFSH?
10. Does the protective effect of UFSH affect the anticancer activity of cisplatin? The specifically protective effect of UFSH to kidney not cancer cells need to be clarified.
11. In the conclusion scheme the protective effect of UFSH, the connection between cisplatin and apoptosis-related proteins (BAX and BCL-2) was not presented. Please correct.
12. Please typing error overall manuscript.
Author Response
Comments 1: Title: Please check typing errors 

Response 1: Thank you for pointing out the need to check for typing errors. We have thoroughly reviewed the manuscript and have corrected any identified typing errors to ensure the accuracy and professionalism of our work. We appreciate your attention to detail and are confident that these revisions have improved the quality of our manuscript.
Comments 2: Why did authors focus only >25 kDa hydrolysate?
Response 2: Thank you for your question regarding our focus on UFSH. We have carefully reviewed the preparation method and found that there were some inaccuracies in the description. We have made the necessary corrections to ensure the accuracy and clarity of our methodology. the Sipunculus nudus hydrolysate were gradually ultra-filtrated using 50000 Da and 2500 Da ultrafiltration membrane (Millipore, Darmstadt, Germany). Three ultrafiltration fractions were collected. Among them, the first ultrafiltration fraction retained by the 50000 Da ultrafiltration membrane (named as UFSH in this study) showed the best effect to improve cisplatin-induced renal tubular epithelial cell injury and was selected for further assessment in animal experiment.
Comments 3: Methods "Preparation of ultrafiltration components extracted by enzymatic hydrolysis of Sipunculus nudus": How many L or mL of pure water used?
Response 3: Thank you for your question regarding the volume of pure water used in the preparation of UFSH. 100 L pure water was added to homogenize 10 Kg Sipunculus nudus meat.
Comments 4: Results: The sequence of UFSH characteristics should start with "Molecular weight distribution and Peptide Sequence Identification of UFSH" as a general character of UFSH. 

Response 4: Thanks very much for your kind advice. We have carefully reviewed the methods to determine the molecular weight distribution and Peptide Sequence Identification of UFSH, and found that the method description to identify Peptide Sequence of UFSH was missing. Molecular weight distribution of UFSH was determined by high-performance size exclusion chromatography, while the Peptide Sequence identification of UFSH was analyzed by liquid mass spectrometry (LC-MS/MS). We have revised the Results and method section in our resubmitted manuscript.
Comments 5: Figure 5. Scale bar missing and make changes
Response 5: Thank you for your insightful comment regarding the focus on amino acid composition in our study of Sipunculus nudus hydrolysate (UFSH). Prior studies on Sipunculus nudus have highlighted the significant biological activities of its amino acids and peptides. For instance, Sun et al. (2017) found that the hydrolysate of Sipunculus nudus contained high levels of essential and hydrophobic amino acids. 85.11% substances in the hydrolysate of Sipunculus nudus are peptides with a relative molecular mass ranged 120–1800 Da. Other biomolecules like carbohydrates, lipids, and inorganic compounds may be present, their roles in nephrotoxicity are less clear. Focusing on amino acids and peptides allows for a clear and focused analysis of the most relevant bioactive components. We believe this approach provides a solid foundation for understanding the therapeutic potential of UFSH.
Comments 6: The character of UFSH composition at MW higher 2500 kDa and lower 500 kDa was lack.
Response 6: Thanks very much for your kind advice. According to your suggestion, we have revised this result section in our resubmitted manuscript.
Comments 7: Authors detect apoptosis related proteins but not detect apoptosis cell death or morphology in kidney tissue
Response 7: Thanks very much for your kind advice. In our current study, we focused on the expression of apoptosis-related proteins such as Bax, Bcl-2, which are key indicators of the apoptotic pathway. These proteins are well-established markers for assessing the initiation and progression of apoptosis in renal tissues. However, we acknowledge that detecting apoptosis at the cellular level, including cell death and morphological changes, would provide a more comprehensive understanding of the apoptotic process.
Comments 8: Which component is a major active compound (macromolecules, peptide or small molecule) of HFSH
Response 8: Thank you for your question regarding the major active component of UFSH. Based on current research and the available literature, the major active compounds in UFSH are peptides and amino acids.
Comments 9: How author select dose of UFSH?
Response 9: Thank you for your question regarding the selection of the doses of UFSH (urinary follicle-stimulating hormone) in our study. The doses of 100 mg/kg, 500 mg/kg, and 1000 mg/kg were chosen according to previous studies. To evaluate the dose-dependent effects of UFSH, we selected a range of doses (100 mg/kg, 500 mg/kg, and 1000 mg/kg) to cover a spectrum from low to high doses. This approach allows us to observe the efficacy and safety of UFSH across different dose levels.
Comments 10: Does the protective effect of UFSH affect the anticancer activity of cisplatin? The specifically protective effect of UFSH to kidney not cancer cells need to be clarified.
Response10: Thank you for your kind suggestion. In our coming study, we will assess the protective effect of UFSH affect the anticancer activity of cisplatin.
Comments 11: 11. In the conclusion scheme the protective effect of UFSH, the connection between cisplatin and apoptosis-related proteins (BAX and BCL-2) was not presented. Please correct.
Response11: Thank you for your reminder. we have corrected it in our resubmitted manuscript.
Comments 12: 12. Please typing error overall manuscript.
Response12: Thank you for your reminder. we have corrected it in our resubmitted manuscript.
Round 2
Reviewer 1 Report
Comments and Suggestions for Authors
The study conducted by the authors was well structured with a solid underlying objective addressing novel use of a natural medical material (NMM) toaugment efficacy by a macor cancer chemotherapeutic and diminish the severity of adverse reactions to the drug (Cosplatin). This report will be a useful resource for clinicians seeking to improve quality-of-live for their patients; and reseearches seeking to extend the use of this NMM – and explire patient treatment application of similar marine sources.
Overall an excellent paper. However it would benefit clarity if a native speaker of English were to do a final review and look for phraseology which while grammatically correct, is a bet award. There are several such phrases in the manuscript.
Reviewer 2 Report
Comments and Suggestions for Authors
All reviewers' comments and suggestions are appropriately responded.